# Interictal Heart Rate Variability as a Biomarker for Comorbid Depressive Disorders among People with Epilepsy

**DOI:** 10.3390/brainsci12050671

**Published:** 2022-05-20

**Authors:** Guliqiemu Aimaier, Kun Qian, Zishuo Zheng, Weifeng Peng, Zhe Zhang, Jing Ding, Xin Wang

**Affiliations:** 1Department of Neurology, Zhongshan Hospital, Fudan University, 180 Fenglin Road, Shanghai 200032, China; 19111210062@fudan.edu.cn (G.A.); peng.weifeng@zs-hospital.sh.cn (W.P.); ding.jing@zs-hospital.sh.cn (J.D.); 2Department of Information and Intelligence Development, Zhongshan Hospital, Fudan University, 180 Fenglin Road, Shanghai 200032, China; qian.kun@zs-hospital.sh.cn; 3State Key Laboratory of Neuroscience, Institute of Neuroscience, Center for Excellence in Brain Science and Intelligence Technology, Chinese Academy of Sciences, Shanghai 200032, China; zszheng2020@ion.ac.cn (Z.Z.); zhezhang@ion.ac.cn (Z.Z.); 4CAS Center for Excellence in Brain Science and Intelligence Technology, Shanghai 200032, China; 5Department of the State Key Laboratory of Medical Neurobiology and MOE Frontiers Center for Brain Science, Institutes of Brain Science, Fudan University, Shanghai 200032, China

**Keywords:** epilepsy, support vector machine, depressive disorders, heart rate variability, autonomic disorders

## Abstract

Depressive disorders are common among people with epilepsy (PwE). We here aimed to report an unbiased automatic classification of epilepsy comorbid depressive disorder cases via training a linear support vector machine (SVM) model using the interictal heart rate variability (HRV) data. One hundred and eighty-six subjects participated in this study. Among all participants, we recorded demographic information, epilepsy states and neuropsychiatric features. For each subject, we performed simultaneous electrocardiography and electroencephalography recordings both in wakefulness and non-rapid eye movement (NREM) sleep stage. Using these data, we systematically explored the full parameter space in order to determine the most effective combinations of data to classify the depression status in PwE. PwE with depressive disorders exhibited significant alterations in HRV parameters, including decreased time domain and nonlinear domain values both in wakefulness and NREM sleep stage compared with without depressive disorders and non-epilepsy controls. Interestingly, PwE without depressive disorder showed the same level of HRV values as the non-epilepsy control subjects. The SVM classification model of PwE depression status achieved a higher classification accuracy with the combination of HRV parameters in wakefulness and NREM sleep stage. Furthermore, the receiver operating characteristic (ROC) curve of the SVM classification model showed a satisfying area under the ROC curve (AUC: 0.758). Intriguingly, we found that the HRV measurements during NREM sleep are particularly important for correct classification, suggesting a mechanistic link between the dysregulation of heart rate during sleep and the development of depressive disorders in PwE. Our classification model may provide an objective measurement to assess the depressive status in PwE.

## 1. Introduction

Depression is one of the most common psychiatric comorbidities in people with epilepsy (PwE), with a prevalence of 23.1% (95% CI, 20.6–28.3%) in previous studies [1]. Importantly, among PwE depression has been associated with poor clinical outcome [2,3], abnormal response to pharmacological and surgical treatment [4] and general low quality of life during recovery [5]. While it remains unclear whether the development of comorbid depression is mechanistically linked with specific processes of epilepsy, there is a consensus for the need to identify the comorbid depression in order to improve the management and prognosis of PwE [1].

A number of screening instruments for depressive disorders are now available for adults However, most of these tools are based on questionnaires such as the Hamilton Depression Scale-17 (HAMD-17) [6], which takes a lot of effort to train specialists to estimate. In addition, diagnosing depression in PwE is challenging due to the potential confounding factors of the atypical features of mood disorders [7]. Therefore, an objective, measurement-based standardized screening for depressive disorders will be of great use to improve the diagnosis and treatment in PwE.

There is currently no measurement-based diagnosis for depression. From a biological point of view, current research regarding the pathogenic mechanisms of epilepsy comorbid depressive disorder focused on neurotransmitter disturbances, immunologic disturbances and genetic factors [8,9,10,11], which are difficult or invasive to test. Previous work suggests that heart rate variability (HRV) parameters might be a promising non-invasive measurement associated with depressive disorders [12]. First, abnormal HRV parameters have been associated with depressive disorders [13,14], with a plausible mechanistic link with autonomic nerve system (ANS) dysfunction [15]. Furthermore, HRV abnormalities have been observed in PwE [16,17,18] and exhibit correlation with seizure severity [19]. However, using HRV measurements as a reliable clinical measurement has been challenging [20]. In this study, we hypothesize that seizure activity may disrupt ANS function in certain PwE, leading to the development of depressive disorders, which can be reflected by the concurrent abnormalities in HRV parameters.

Following this hypothesis, we explored the correlation between HRV abnormalities and the development of depressive disorders in PwE in this study. We found decreased HRV parameters in PwE compared with non-epilepsy controls and the alteration was significant in PwE with depressive disorders. Classification of PwE depression status was achieved through a linear support vector machine (SVM) based on interictal HRV parameters.

## 2. Materials and Methods

### 2.1. Study Design and Patient Selection

Both PwE and non-epilepsy controls were consecutively obtained from subjects who underwent 2 hours video electroencephalography (EEG) from November 2020 to June 2021. All participants underwent an EEG exam at the neurology department of the Zhongshan Hospital, Fudan University, Shanghai, China. The neuropsychological assessment was performed right after the EEG exam.

A subject was included in the PwE group: (1) if he or she was aged between 18 and 75 years, (2) if he or she had received a diagnosis of epilepsy according to the International League Against Epilepsy (ILAE) classification of epilepsies (2017) based on clinical data and EEG recording [21], (3) if he or she does not have cardiovascular related disease, (4) if he or she did not take antidepressant medications before [22], (5) if he or she does not have systematic disease such as autoimmune illnesses, hematological disorders, infectious diseases, thyroid related illnesses, pregnancy or breast-feeding, (6)if he or she had a Mini Mental State Examination (MMSE) score greater than or equal to 22, (7) if he or she had a General Anxiety Disorder-7 (GAD-7) score lower than 5 and (8) if he or she agreed with and signed a written informed consent form approved by the ethics committee of Zhongshan Hospital (Figure 1).

A subject was included in the non-epilepsy controls group as the same condition as the PwE group, but who does not have any epileptic disorders before and the result of EEG was normal.

Considering the psychiatric side effect of antiepileptic drugs (AEDs) in PwE [4], whether taking AEDs and the number of AED types at the time of examination have also been recorded. Levetiracetam (LEV) was considered as one of the most mood-affecting AEDs in the relevant study [23], thus we recorded whether the PwE was taking LEV. Finally, the clinical features were collected as follows: patient’s age, gender, disease history, physical and neurological examination results, duration of epilepsy, number of self-reported seizures in the previous 1 year, seizure origin, whether taking AEDs, number of AED types and whether taking LEV. The seizure types and epilepsy syndromes were determined according to the ILAE classification of epileptic seizures [21].

### 2.2. Diagnosis of Epilepsy and Video-EEG Evaluation

#### 2.2.1. Diagnosis of Epilepsy

All PwE were evaluated according to the diagnostic protocols at the Neurology Department of Zhongshan Hospital. These protocols included a clinical examination by trained neurologists. Additionally, additional complementary measures including video-EEG monitoring, magnetic resonance image (MRI) and neuropsychological and psychiatric assessment were performed to confirm the epilepsy origin of the epileptogenic zone.

#### 2.2.2. Video-EEG Evaluation

All subjects included in this study underwent 2 hours video-EEG monitoring in a continuous supine position. For the EEG monitoring, a Natus-Xltek EMU 128FS machine was used and the EEG was sampled at 250-Hz. A single channel of ECG, 4 channels of EMG and 25 channels of EEG were synchronously recorded and exported in European Data Format (EDF). Referential montages (A1, A2) were used for analysis. All interictal recordings were obtained using the international 10–20 system.

#### 2.2.3. Sleep Recording

EEG data were acquired using a 25-channel EEG with 4-channel EMG electrodes placed according to the international 10–20 System. Raw data were visually scored into wakefulness and non-rapid eye movement (NREM) sleep stage according to the Rechtschaffen and Kales manual [24]. After scoring the wakefulness and NREM sleep stages, three consecutive 5-min period epochs without artifacts and arousals in each stage were selected.

### 2.3. Neuropsychological Assessment

All subjects included in this study underwent a complete neuropsychological assessment after the EEG monitoring. Neuropsychological assessment was performed by one trained specialist according to a standardized protocol especially designed for PwE [6,25]. Neuropsychological history was obtained from each patient and relatives, complemented by information from families. The MMSE was performed to evaluate subjects’ cognitive function, which could rapidly screen the participants with severe cognitive impairments. If the MMSE score was lower than 22, participants were excluded from the research. The Chinese version of HAMD-17 and MMSE scales have both been tested and proved to have good validity and reliability [26,27]. HAMD-17 was performed to recognize depressive symptoms and to evaluate the severity of the depression. The seventeen questions of the HAMD reflect several aspects of depressive disorders, including depressed mood, feelings of guilt, suicidal thoughts, insomnia, anhedonia, psychomotor retardation, agitation, anxiety, somatic symptoms, sexual interest, hypochondriasis, loss of weight and insight. According to the grading standard, scores of 0–6 are considered as without depression and scores ≥7 as with depression in PwE [25]. GAD-7 was performed for excluding the influence of other mood disorders such as anxiety, and those with a score equal or greater than 5 were also excluded.

### 2.4. HRV Analysis

Calculation of HRV parameters was carried out with an open-source Matlab toolbox for analyzing HRV (MarcusVollmer-HRV-1b). Considering the possible effects of wakefulness and sleep stages on HRV parameters [28], the indices were analyzed in both stages. Three consecutive 5-min period ECG data without artifacts and arousals from wakefulness and NREM sleep stage were analyzed, respectively, and an average value of every index from the three periods were calculated.

The common HRV measures include time domain, frequency domain and nonlinear domain measure:

Time domain measures: SDNN (standard deviation of all RR intervals), RMSSD (root mean square of the difference of adjacent RR intervals) and PNN50 (probability of RR intervals greater than 50 msec). SDNN represents a global measure of HRV and provides information about all HRV components. RMSSD is considered a powerful measure of high frequency power (HF, 0.15–0.4 Hz) variations in short-term recording, as it provides a useful evaluation of HF and vagal tone [29]. Given the dependence of SDNN on record length, SDNN values should be compared to SDNN obtained from recordings with similar duration. PNN50 is a percentage of consecutive intervals that differ by more than 50 msec, which reflects fast high frequency variability changes [30].

Frequency domain measures: average low frequency power (LF, 0.04–0.15 Hz) and high frequency power (HF, 0.15–0.4 Hz)—were obtained by time averaging for each subject and each condition. LF reflects the modulations of the sympathetic and the parasympathetic nervous system, whereas HF mainly reflects parasympathetic nervous system activity [31]. The LF/HF ratio expresses the balance between sympathetic and parasympathetic nervous system activity [32].

Nonlinear domain measures: we calculated the geometric nonlinear measures and entropies nonlinear measures. Geometric nonlinear methods such as Poincaré plots can be determined by ellipse width SD1 (associated with rapid variations between heart beats) and ellipse length SD2 (corresponds to long-term variability of RR intervals). The SD1/SD2 reflects the relationship between short- and long-term HRV [30]. Approximate entropy (ApEn) as a nonlinear entropy method for analyzing HRV, could determine the degree of irregularity of the full length of RR time series. The low entropy values are characteristic of regular time series, while higher values are inherent in stochastic data [33].

### 2.5. EEG Power Spectral Density Analysis

For subsequent analyses of EEG Power Spectral density (PSD) data as a supplement to the HRV measurements in classifying the depression status, average power estimates were extracted for each of the below frequency band: α (8–13 Hz), β (14–30 Hz), θ (4–7.5 Hz) and δ (0.3–3.5 Hz) from the same continuous 5-min period as ECG data. In order to assess the spatial distribution of the effects, the average of PSD values of above four frequency bands were calculated separately for each of the 25 channels.

### 2.6. Support Vector Machine (SVM) Classification

Combinations of HRV parameters as features and binarized HAMD-17 (≥7 or <7, represents for with or without depression) as the response were fed to a SVM classification algorithm [34]. SVM classifications were conducted using mapping features onto a violin plot, which exhibited the accuracy of classification between observations of combinations of HRV parameters. The SVM classifiers were configured using C-classification with the radial basis kernel function and five-fold cross validation. Thus, we trained 80% of the participants with complete data and tested classifier performance using the remaining 20%. The units of the two variables (with or without depression) were different, therefore the value ranges were normalized to range between 0 and 1 in order to eliminate any bias from the differences in the units. Then, we plotted the SVM decision boundary in a principal component analysis (PCA) space, a receiver operating characteristic (ROC) curve for model prediction, and evaluated the classifier’s performance using the confusion matrix (the number of correctly classified observations per total observations). SVM classifications were performed with Python 8 [35].

### 2.7. Statistical Analysis

Descriptive statistics and outcome measures for each item were shown as mean ± standard deviation (SD). Differences between unrelated groups were assessed with the non-parametric Mann–Whitney U test and non-parametric Kruskal–Wallis test was used for more than two groups. If the Kruskal–Wallis test was significant, post hoc analysis was performed using the Bonferroni correction test to analyze one-to-one differences between groups. Chi-square test was used to discriminate categorical variables. Spearman correlation analyses were used to evaluate the relationships between scores of HAMD and the clinical variables. A post hoc power analysis was performed based on the subjects recruited in the study. All the statistical analyses were performed using GraphPad Prism 8. All the *p*-values were <0.05 at the two tails considered to indicate statistical significance.

## 3. Results

### 3.1. Clinical Characteristics of the Subjects

A total of 186 subjects were recruited in this study, out of whom 28 subjects were excluded as they did not meet the inclusion criteria (Figure 1). Sixty PwE with HAMD-17 score greater than 7 were recruited in the PwE with depression group, the 51 matched PwE with HAMD-17 score ranged 0–6 were set as the PwE without depression group, and the 47 matched healthy volunteers without epileptic and depressive symptoms were included in the non-epilepsy control group. The groups were matched for age and gender (*p* = 0.099 and *p* = 0.630, respectively; Table 1). A post hoc power analysis was performed based on the samples recruited in the study. Found 80% power (alpha = 0.05, two-tailed) was provided among non-epilepsy controls and PwE with and without depression group in an ANOVA test, while 81.7% power (alpha = 0.05, two-tailed) was provided between PwE with and without depression group in a t-test for independent samples.

To estimate the possible role of epilepsy related features in the presence of depression, a comparison of clinical characteristics between PwE with and without depression groups was performed. While both groups exhibited similar patterns of seizure origins and duration, PwE with depression status showed the higher frequency of epileptic sessions (Table 2). In addition, we found cognitive deficit in PwE with comorbid depressive symptoms. (Table 2). We did not find a significant difference in the AEDs usage between groups.

To further clarify the correlation between the relevant variables and depressive status, a Spearman correlation analysis was performed between the HAMD-17 score and variables such as gender, age, epilepsy origin and severity, AEDs and MMSE score. The results showed that only the MMSE score (r = −0.308, *p* = 0.001) negatively correlated with HAMD, which indicates that a decrease in cognitive function might be associated with depressive disorders in PwE.

### 3.2. Abnormal Heart Rates Are Associated with the Comorbid Depression in PwE

We used a series of derivatives from the raw ECG data to evaluate HRV. Specifically, we measured the intervals between the R-peaks on the ECG traces (Figure 2A) and calculated their mean heart rate (HR), standard deviation (SDNN), root mean square (RMSSD) and probability of greater than 50 msec (PNN50). We further applied a band pass filter on the raw ECG data and took the low frequency power (LF, 0.04–0.15 Hz), high frequency power (HF, 0.15–0.4 Hz) and their ratio. Finally, we calculated the nonlinear indexes such as approximate entropy (ApEn) and Poincaré plots (SD1, SD2 and ratio of SD1/SD2), which detect regularity of RRI time series and relationship between short- and long-term HRV, respectively.

Preliminary analysis showed that in both the control and the patient group, NREM sleep was associated with longer R-peak intervals and increased variability compared with wake periods (Figure A1). Thus, we compared each period separately between the control and the patient group. We found that compared with the control group, PwE showed decreased time domain (SDNN 38.98 ± 1.56 msec vs. 46.88 ± 2.85 msec, *p* = 0.006; RMSSD 28.13 ± 1.33 msec vs. 32.03 ± 2.17 msec, *p* = 0.0449; PNN50 9.4 ± 1.15 vs. 10.65 ± 1.63, *p* = 0.0332) and nonlinear domain variability (SD1 20.4 ± 10.6 vs. 22.9 ± 9.9, *p* = 0.0453; SD2 49.4 ± 22.0 vs. 57.1 ± 19.9, *p* = 0.0167) during the wakefulness stage. No significant difference has been found in the NREM sleep stage (Figure 2B).

Furthermore, when separating PwE with depression status we found similar changes in HRV parameters and to an even greater extent (Table 3). Post-hoc analysis was performed using the Bonferroni correction to analyze differences between groups and found that PwE with depression had a significant decrease of time domain (SDNN, RMSSD and PNN50, all of the *p* < 0.05 after Bonferroni corrections) and nonlinear domain (SD1 and SD2, both *p* < 0.05 in wakefulness stage and SD1 *p* < 0.05 in NREM sleep stage after Bonferroni corrections) parameters compared with PwE without depression and non-epilepsy controls (Table 3). PwE without depression showed almost the same level as non-epilepsy controls in each of the HRV parameters (all of the *p* > 0.05 after Bonferroni corrections) (Table 3). These data indicate that the HRV measurements can be indicators that correlate with the patients’ epilepsy and comorbid depression states.

### 3.3. HRV Parameters, Particularly during NREM Sleep, Are Capable of Classifying PwE with and without Depression

Following the intuition derived from single parameter analysis above, we next explored the possibility that combining these parameters may provide an accurate classification of the PwE depression states. To this end, we first plotted the global variance of the data using the t-distributed stochastic neighborhood embedding (t-SNE). We found that the comorbid depression did not show good separation on the t-SNE space (Figure 3A), indicating that this information was not the main contributor to the variance in the dataset; therefore, we then utilized a supervised method in the following analysis.

We constructed the SVM model using HRV parameters as features and binarized HAMD-17 as the response. For feature selection, the limited patient numbers prevented us from using a strong L1 regularization with proper cross validation. Thus, we trained and tested models with all possible combinations of HRV parameters with 54/13 for training and testing group. We found that models taking four HRV parameters showed the best performance (Figure 3B), with the parameters of the best model (accuracy 67.3%) being HR and HF in wakefulness and HR and ApEn in NREM sleep. However, lower accuracy (65.2%) was achieved when the nonlinear Poincaré indexes replaced ApEn in the linear SVM classification model. Interestingly, we found that HRV parameters during NREM sleep (up to 65.5% accuracy) were more informative for comorbid depression compared with those during wakefulness (up to 59.9% accuracy) (Figure 3C). We here showcased the performance in our best model. We plotted the SVM decision boundary in a PCA space (Figure 3D), a ROC curve for model prediction (with an area under the ROC curve 0.758) (Figure 3E) and its prediction confusion matrix (Figure 3F), all demonstrating reasonable performance of the model. These data indicate that HRV parameters reflect the comorbid depression status in PwE.

### 3.4. Combining HRV and EEG PSD Data Further Improves Depression Classification

We further explored measurements from electroencephalography (EEG) as a supplement to the HRV measurements in classifying the depression status in SVM classifier. EEG data were acquired with surface electrodes and transformed into an average PSD at α (8–13 Hz), β (14–30 Hz), θ (4–7.5 Hz) and δ (0.3–3.5 Hz) frequencies. Similar to HRV parameters, PSD measurements showed abnormal patterns between the non-epilepsy control group and PwE (Figure A2) and the abnormality was more prominent in PwE with comorbid depression (Figure 4A).

Indeed, we found a higher accuracy (83%) in depression status classification when we combined the HRV parameters with PSD values in the model. Still, using a PCA projection of the decision boundary (Figure 4B) and the confusion matrix (Figure 4C) of the model showcased the superior classification of the combined model. On average, models including the EEG PSD measurements showed better performance than the models using HRV parameters only (Figure 4D), indicating that EEG PSD measurements contained additional information regarding the PwE depressive status.

## 4. Discussion

This is the first report focusing on identifying PwE comorbid depressive disorder with interictal HRV measurements. We found better reproducibility of accuracy in the classifier of PwE depression status on the combination of four parameters: HR and HF in wakefulness stage and HR and ApEn in NREM sleep stage.

This present study confirms that PwE have impaired cardiac autonomic regulation with reduced HRV time domain and nonlinear domain parameters, and it was more significant in PwE with depressive disorders. The HRV parameter in PwE without depressive disorders showed the same level as non-epilepsy controls. Furthermore, the classifying analysis revealed that HRV measurements could be used as a noninvasive and objective classification indicator for PwE with or without depressive disorders in a stable reproducibility.

HRV is a noninvasive indicator of ANS that is calculated by the variation in the time interval between consecutive heartbeats [29] (Figure 2A). Evaluating HRV is useful from a physiologic perspective, because HRV reflects the balance between parasympathetic and sympathetic activity in the ANS [32]. In general, increases in HRV parameters reflect a shift toward parasympathetic dominance, whereas decreases indicate relative increases in sympathetic activity [36]. Spectral analysis provides information on how the power of HRV is distributed as a function of frequency. The total power of RR intervals corresponds to the sum of LF and HF, while vagal tone is considered to be a major contributor to the HF component and the LF has been believed to reflect both sympathetic and vagal influence. Sympathovagal balance is frequently described by LF/HF ratio and showed a slight predominance of LF over HF during resting conditions in healthy subjects [32]. The lower values of nonlinear measures, such as SD1 and SD2 in patients, also indicate the dysfunction of ANS and are used as a prognostic marker for the presence of cardiovascular disease [30]. We found altered time domain and nonlinear domain HRV parameters in PwE compared with non-epilepsy controls and the alteration was more significant in those with depressive disorders, suggesting a mechanistic link between the dysregulation of ANS in PwE leading to the depressive disorders.

A bidirectional relation between depressive disorders and epilepsy has been suggested by several population-based studies and is supported by experimental studies [37,38], however the underlying etiology is still illusive. The potential pathogenic mechanisms operant in both disorders may explain such a relation. These mechanisms include a hyperactive hypothalamic–pituitary–adrenal (HPA) axis and its neuroanatomic and neuropathologic complications, disturbances in serotonergic, noradrenergic, γ-aminobutyric acid GABAergic and glutamatergic neurotransmitter systems and genetic factors, all of which may be interrelated [39]. Alterations of ANS functioning that promotes vagal withdrawal are reflected in reductions of HRV indexes and characterizes emotional dysregulation, decreased psychological flexibility and defective social engagement, which in turn are linked to prefrontal cortex hypoactivity [13]. Under stress conditions, specific areas of the prefrontal cortex become hypoactive, which implies disinhibition of sympathoexcitatory circuits and energy mobilization [40]. Through the findings in this study, we propose a hypothesis that the altered functional network connectivity in PwE leads to an alteration of ANS functioning, which contributes to resulting depression comorbidity of epilepsy. The mechanism-related studies are needed to be further improved.

The diagnosis of epilepsy associated depression has been performed by questionnaire instruments such as HAMD-17 for a long time. However, the diagnosis of epilepsy associated depression has a lack of consistency [7]. In this study, we found a new short-term, invasive and objective instrument that could be utilized to classify and assist diagnosis of epilepsy associated depression. We believe that this is the first study to classify epilepsy comorbid depression with HRV parameters.

## 5. Conclusions

In this study, a linear SVM classifier based on HRV parameters is proposed and applied to PwE depression status classification and provides a new objective indicator for the identification of epilepsy comorbid depression. Comparisons of HRV parameters between groups have been performed and found significant decreases of time- and nonlinear domain measures in PwE with depression, reflecting a shift toward sympathetic dominance. That may suggest a new underlying mechanism for the development of depressive disorders in PwE that autonomic dysfunction might contribute to comorbid depression in epilepsy. Moreover, we found that HRV measurements during NREM sleep are particularly important for correct classification, suggesting a mechanistic link between the dysregulation of heart rate during sleep and the development of depressive disorders in PwE.

Nevertheless, our work has some limitations. The study has a cross-sectional design and there was no further follow-up. However, there is sufficient evidence to suggest a close relationship between HRV alteration and PwE depression status. The primary cause of sympathetic dysfunction in epilepsy and epilepsy comorbid depression remains to be defined. Therefore, longitudinal large-scale studies are now demanded to assess the natural course of cardiac autonomic dysfunction in PwE. We did not use a ‘gold standard’ for the HRV measurements (i.e., 24 h Holter ECG recorders); however, we used a short-term protocol, involving 5-min measurements in wakefulness and NREM sleep stage, recommended in studies of HRV in epilepsy and considered as a standardized minimum protocol [20].

## Figures and Tables

**Figure 1 brainsci-12-00671-f001:**
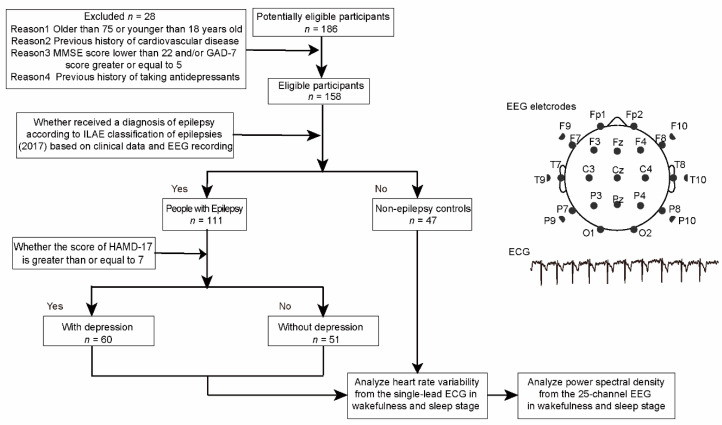
Flowchart of the study. Workflow for subjects included in the study (on the left) and channel locations of EEG and representative ECG trace (on the right). HAMD-17—17-item Hamilton Depression Scale; MMSE—Mini Mental State Examination; GAD-7—Generalized Anxiety Disorder-7; ECG—electrocardiography; EEG—electroencephalography.

**Figure 2 brainsci-12-00671-f002:**
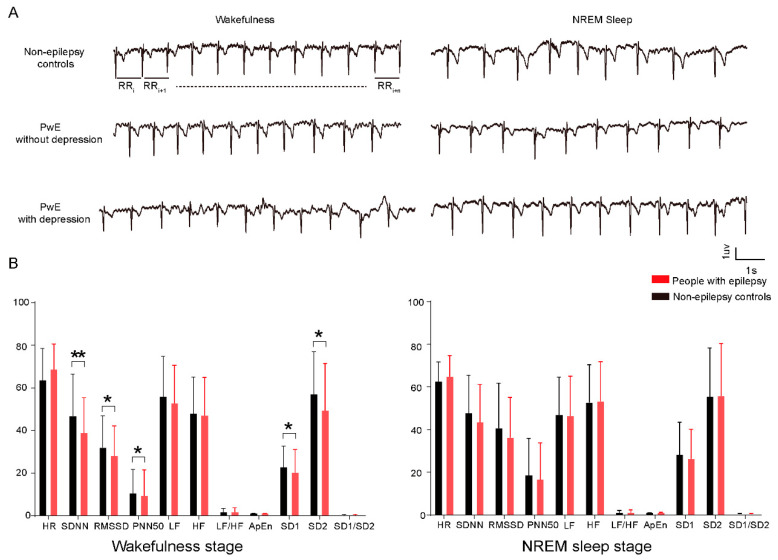
Significant differences in HRV parameters have been observed between people with epilepsy (PwE) and non-epilepsy controls. (**A**) Representative electrocardiography RR intervals in non-epilepsy controls and PwE with and without depression. (**B**) In comparison with the non-epilepsy controls, PwE exhibited a lower time domain and nonlinear domain indexes in wakefulness stage (Mann–Whitney test, SDNN *p* = 0.006, RMSSD *p* = 0.0449, PNN50 *p* = 0.0332, SD1 *p* = 0.0453, SD2 *p* = 0.0167); no significance differences have been observed in NREM sleep stage. * *p* < 0.05, ** *p* < 0.001. Concepts of HRV parameters in the figure have been explained in detail in Section 2.

**Figure 3 brainsci-12-00671-f003:**
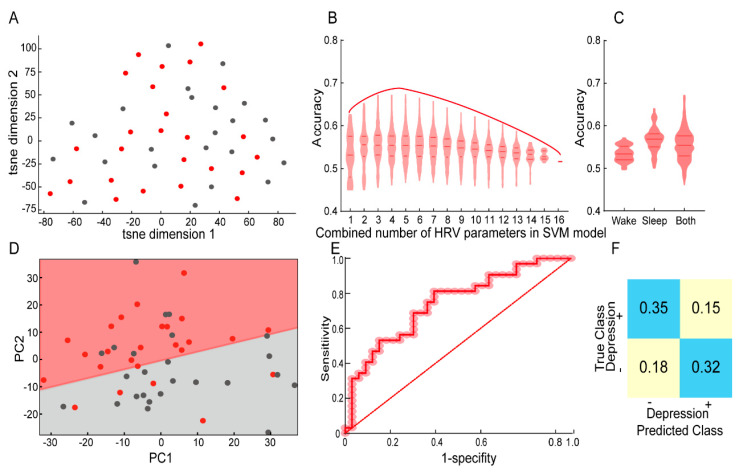
HRV parameters, particularly during NREM sleep, are capable of classifying PwE with and without depression. (**A**) A t−distributed stochastic neighborhood embedding (t−SNE) for classifying PwE with/without depression based on HRV parameters. (**B**) Computing the accuracy of all possible combinations of HRV parameters through a linear support vector machine (SVM) model, high degree classification accuracy distributed mainly in the combination of 4 parameters. (**C**) Computing accuracy in wakefulness and NREM sleep stage, highest accuracy was achieved while parameters involved both wakefulness and NREM sleep stage. (**D**) SVM decision boundary showed in a principal component analysis (PCA) space. (**E**) ROC curve for the model accuracy prediction, the model accuracy estimate fit the data at an acceptable level (Hosmer−Lemeshow test, *p* = 0.5676) and the area under the ROC curve was 0.7576. (**F**) Prediction confusion matrix for the selected optimal model. The red and gray dots in (**A**,**D**) representing PwE with and without depression state, respectively. PwE−people with epilepsy.

**Figure 4 brainsci-12-00671-f004:**
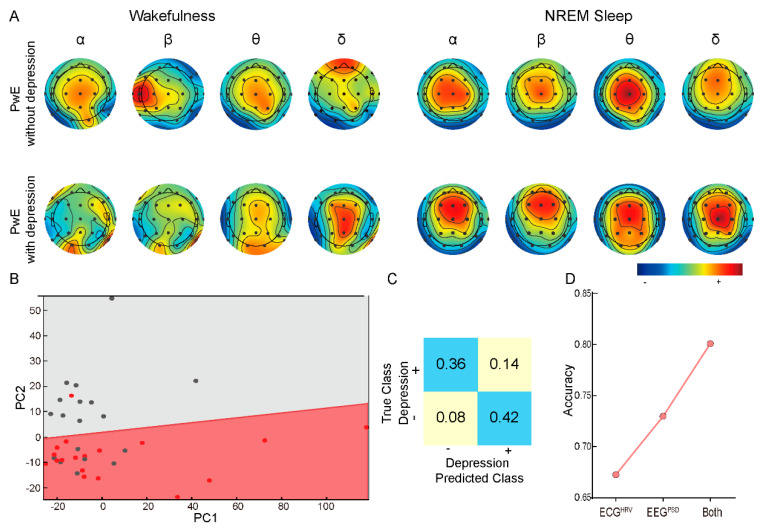
Combining HRV and EEG power spectral density (PSD) data further improves depression classification. (**A**) Topographical representation of four frequency band (α (8−13 Hz), β (14−30 Hz), θ (4−7.5 Hz), δ (0.3−3.5 Hz)) in PwE with/without depression. (**B**) Principal component analysis (PCA) has been performed through selected HRV parameters and EEG PSD values, showing a better classification than HRV parameters only. (**C**) Confusion matrix analysis through selected HRV and EEG PSD data showed lower false positives/negatives and higher true positives/negatives than HRV parameters only. (**D**) Comparison of classification accuracy in HRV and EEG PSD data, the highest accuracy was achieved while combining HRV and EEG PSD data in the algorithm. The red and gray dots representing PwE with and without depression in Figure 4B, respectively. PwE−people with epilepsy.

**Table 1 brainsci-12-00671-t001:** Comparisons of the demographic characteristics among PwE with depression, PwE without depression and non-epilepsy controls.

	*n*	Gender (F/M)	*p*	Age (Year, Mean ± SD)	*p*
PwE with depression	60	32/28	0.630	50.17 ± 15.23	0.099
PwE without depression	51	23/28		43.69 ± 17.6	
Non-epilepsy controls	47	25/22		49.79 ± 16	

Kruskal–Wallis test for continuous variables (Age) and Chi-square test for categorical variables (Gender) have been used to discriminate among groups. PwE—people with epilepsy; SD—Standard Deviation.

**Table 2 brainsci-12-00671-t002:** Comparisons of clinical characteristics between PwE with and without depression groups.

Variables		Without Depression (*n* = 51)	With Depression (*n* = 60)	*p*-Value
**Seizure characteristics**	Seizure Origin			0.486
	Temporal lobe	27	37	
	Frontal lobe	5	6	
	Occipital lobe	1	3	
	Unknown origin	18	14	
	Mean duration of disease (Year, mean ± SD)	7.65 ± 9.7	10.03 ± 14.5	0.652
	Seizure frequency			0.035
	per week	1	7	
	per month	16	8	
	per year	23	34	
	no seizure in a year	11	11	
**AEDs information**	Taking AEDs	46/51	53/60	0.753
	Number of AEDs (mean ± SD)	1.24 ± 0.86	1.23 ± 0.83	0.677
	LEV	14/51	18/60	0.835
**MMSE score (mean ± SD)**		28.3 ± 1.9	27.2 ± 2.3	0.004

Chi-square test was performed to compare categorical data such as gender, seizure origin, seizure frequency, whether taking anti-epileptic drugs (AEDs) and whether taking levetiracetam (LEV); seizure frequency has been classified as at least one attack per week/per month/per year or no episode in previous one year and the chi-square test was performed to examine the distribution of PwE in above four frequency range. Mann–Whitney test was performed to compare continuous data such as age, seizure duration, number of AEDs and MMSE score. PwE—people with epilepsy; MMSE—Mini Mental State Examination; SD—Standard Deviation.

**Table 3 brainsci-12-00671-t003:** Comparison of HRV parameters between PwE and non-epilepsy controls.

Stage	HRV Parameters	Non-Epilepsy Controls (*n* = 47)	PwE without Depression (*n* = 51)	PwE with Depression (*n* = 60)	*p*-Value	Post-Hoc (Bonferroni Corrections)
Wakefulness stage	HR	66.4 ± 9.7	67.5 ± 9.6	71.3 ± 10.3	0.0421	all *p*’s > 0.05
	SDNN (msec.)	46.9 ± 19.6	44.4 ± 17.3	34.4 ± 14.3	<0.0001	*p*2, *p*3 < 0.001, *p*1 = 1
	RMSSD (msec.)	32.0 ± 14.8	32.0 ± 14.5	24.8 ± 12.9	0.0006	*p*2, *p*3 < 0.05, *p*1 = 1
	PNN50	10.7 ± 11.2	12.7 ± 12.6	6.6 ± 11.0	0.0001	*p*2, *p*3 < 0.05, *p*1 = 1
	LF	55.9 ± 18.9	51.8 ± 16.6	53.7 ± 18.7	0.4602	all *p* > 0.05
	HF	48.0 ± 17.2	48.1 ± 16.6	46.3 ± 18.8	0.6197	all *p* > 0.05
	LF/HF	1.8 ± 1.6	1.7 ± 1.7	1.9 ± 1.9	0.6242	all *p* > 0.05
	ApEn	1.1 ± 0.2	1.1 ± 0.1	1.1 ± 0.1	0.8226	all *p* > 0.05
	SD1	22.8 ± 9.9	23.3 ± 10.8	17.9 ± 9.9	0.004	*p*2, *p*3 < 0.05, *p*1 = 1
	SD2	57.1 ± 19.9	56.4 ± 23.6	43.5 ± 18.8	0.003	*p*2, *p*3 < 0.05, *p*1 = 1
	SD1/SD2	0.42 ± 0.15	0.44 ± 0.19	0.43 ± 0.21	0.823	all *p*’s > 0.05
Non-rapid eye movement (NREM) sleep stage	HR	62.8 ± 9.0	62.5 ± 9.6	67.9 ± 9.5	0.0186	*p*2 < 0.05, *p*1, *p*3 > 0.05
	SDNN (msec.)	47.9 ± 17.7	48.4 ± 18.0	38.9 ± 16.3	0.0153	*p*2, *p*3 < 0.05, *p*1 = 1
	RMSSD (msec.)	40.8 ± 20.9	41.3 ± 20.4	31.6 ± 15.9	0.0180	*p*2, *p*3 < 0.05, *p*1 = 1
	PNN50	18.7 ± 17.1	21.6 ± 18.8	11.8 ± 14.1	0.0107	*p*2, *p*3 < 0.05, *p*1 = 1
	LF	47.1 ± 17.6	45.1 ± 19.2	47.9 ± 17.9	0.6908	all *p* > 0.05
	HF	52.9 ± 17.6	54.9 ± 19.2	52.0 ± 17.9	0.6864	all *p* > 0.05
	LF/HF	1.2 ± 0.9	1.2 ± 1.2	1.2 ± 1.0	0.6901	all *p* > 0.05
	ApEn	1.1 ± 0.1	1.1 ± 0.1	1.1 ± 0.1	0.4881	all *p* > 0.05
	SD1	28.34 ± 15.1	29.4 ± 14.6	23.4 ± 12.6	0.0396	*p*2 < 0.05, *p*1, *p*3 > 0.05
	SD2	55.5 ± 22.9	61.3 ± 24.9	50.6 ± 23.4	0.1121	all *p* > 0.05
	SD1/SD2	0.53 ± 0.19	0.51 ± 0.19	0.47 ± 0.17	0.4673	all *p* > 0.05

Kruskal–Wallis test was performed to compare the HRV parameters among non-epilepsy controls, PwE with and without depression. Post-hoc analysis was performed using the Bonferroni correction test (*p*1 non-epilepsy controls vs. PwE without depression; *p*2 PwE without depression vs. PwE with depression; *p*3 non-epilepsy controls vs. PwE with depression). Significance differences have been found both in wakefulness and NREM sleep stage in PwE with depression vs. without depression and PwE with depression vs. non-epilepsy controls. No significance difference has been found between PwE without depression and non-epilepsy controls. PwE—people with epilepsy; SD—Standard Deviation. Concepts of HRV parameters in table have been explained in detail in Section 2.

## Data Availability

The raw data supporting the conclusions of this article will be made available by the authors, without undue reservation.

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
