# Peer review of "Interictal Heart Rate Variability as a Biomarker for Comorbid Depressive Disorders among People with Epilepsy"

_brainsci, 2022, doi:10.3390/brainsci12050671_

Round 1

Reviewer 1 Report

Comments and Suggestions for Authors

In this paper, the authors are proposing the skills of automated classification of epilepsy comorbid depressive disorders from heart rate variability. In this study, the authors hypothesized that seizure activity may disrupt ANS function in certain patients, leading to the development of depressive disorders, which can be reflected by the concurrent abnormalities in HRV parameters. As we know, SDNN, RMSSD, and PNN50 are time-domain parameters of HRV. Nonlinear domain parameters of HRV (e.g., small scale multiscale entropy index (not ApEn)) were not included in the manuscript.

Although I find the result a little interesting, I have significant reservations about some technical and scientific aspects of the current study:

Major issues

  1. The title of the manuscript (e.g., general statement) needs to be modified to fit the “new” in your study compared with others (model, design, idea, application, findings, conclusions?).
  2. In lines 70-71, the result statement “An accuracy (67.3%) for classification of epileptic patients with or without depression disorders achieved through the combination of HRV parameters.” should not be included in the introduction section.
  3. The authors must also describe the methods in detail. Please add a small scale multiscale entropy index or multiscale Poincaré index (refer to PMID: 30415711 DOI: 10.1016/j.cmpb.2018.10.001) to replace ApEn.
  4. The words in the Conclusions section need to be rewritten.

Minor issues

  1. In general, p values are set as: ** p < 0.001, * p < 0.05, however *** p < 0.001, **** p < 0.0001 are not appropriate in the figures and tables.
  2. I suggest that the authors carefully go through the paper again and correct all typos.

Reviewer 2 Report

The abstract needs modification. The HRV is a higher frequency signal and EEG for NREM and sleep stage are low frequency predominant signals how they will match with PSD analysis. The depression and HRV may be resulted due to some other medical problem also. Kindly progress and ascertain in this direction. The methods are good and presentation is very poor. The conclusion needs modification. figure 2 needs more explanation about the concepts.

Reviewer 3 Report

The paper reports an interesting approach to the classification of depressive disorders in epilepsy patients. The study applied an interesting methodology approach to evaluate a relevant clinical topic. The paper is well structured and the data are well presented. Also, graphical representations are very useful. However, I have some comments for the authors that could help them to clarify their manuscript.

  • Introduction, lines 53-55: you presented the previous literature as if there is a different pathway for depression in the general population and in patients with epilepsy. Is it true? I think you should revise this part.
  • Have you performed a prior power analysis? 
  • Have you evaluated the possibility to apply a correction for multiple analyses?
  • Have you evaluated the possible role of the frequency of the seizure episodes in the presence of depression?
  • ANOVA is a parametric analysis. Please change the name in the text. If you used a different analysis than ANOVA you should report its name. 
  • Table 3, I think you report twice the p-values. In the ANOVA results the data seems to be p-values but they have also * for significance. It is redundant and for clarity you should report the value of the statistical analysis in that column. (I think the same is valid for the other tables).

Round 2

Reviewer 2 Report

All the corrections are included in the paper. Hence, the paper may be accepted.

Author Response

Thank you for your comments and approval. Best regards.

Reviewer 3 Report

I think the authors have performed a robust revision of the manuscript. The paper is acceptable in the present form in my opinion. Thank you again for the opportunity to review this interesting manuscript. 

Author Response

(The authors gave the same response as above.)
